# *De Novo* Purine Nucleotide Biosynthesis Pathway Is Required for Development and Pathogenicity in *Magnaporthe oryzae*

**DOI:** 10.3390/jof8090915

**Published:** 2022-08-29

**Authors:** Meng-Yu Liu, Li-Xiao Sun, Hui Qian, Yun-Ran Zhang, Xue-Ming Zhu, Lin Li, Shuang Liang, Jian-Ping Lu, Fu-Cheng Lin, Xiao-Hong Liu

**Affiliations:** 1State Key Laboratory of Rice Biology, Institute of Biotechnology, Zhejiang University, Hangzhou 310058, China; 2State Key Laboratory for Managing Biotic and Chemical Threats to the Quality and Safety of Agro-Products, Institute of Plant Protection and Microbiology, Zhejiang Academy of Agricultural Sciences, Hangzhou 310021, China; 3College of Life Science, Zhejiang University, Hangzhou 310058, China

**Keywords:** *Magnaporthe oryzae*, *de novo* purine biosynthesis, MoAde8, fungal growth, TOR activity

## Abstract

Purine nucleotides are indispensable compounds for many organisms and participate in basic vital activities such as heredity, development, and growth. Blocking of purine nucleotide biosynthesis may inhibit proliferation and development and is commonly used in cancer therapy. However, the function of the purine nucleotide biosynthesis pathway in the pathogenic fungus *Magnaporthe oryzae* is not clear. In this study, we focused on the *de novo* purine biosynthesis (DNPB) pathway and characterized MoAde8, a phosphoribosylglycinamide formyltransferase, catalyzing the third step of the DNPB pathway in *M. oryzae*. MoAde8 was knocked out, and the mutant (∆*Moade8*) exhibited purine auxotroph, defects in aerial hyphal growth, conidiation, and pathogenicity, and was more sensitive to hyperosmotic stress and oxidative stress. Moreover, ∆*Moade8* caused decreased activity of MoTor kinase due to blocked purine nucleotide synthesis. The autophagy level was also impaired in ∆*Moade8*. Additionally, MoAde5, 7, 6, and 12, which are involved in *de novo* purine nucleotide biosynthesis, were also analyzed, and the mutants showed defects similar to the defects of ∆*Moade8*. In summary, *de novo* purine nucleotide biosynthesis is essential for conidiation, development, and pathogenicity in *M. oryzae*.

## 1. Introduction

Rice blast, prone to occur in all rice-growing regions around the world, causes a substantial reduction in rice production. *Magnaporthe oryzae*, the agent of rice blast, is a filamentous ascomycete fungus, which can attack rice cells at all developmental stages [1]. Conidiation is an essential phase in the disease cycle and epidemic transmission of *M. oryzae*. After conidium attaches to rice plants, it starts to recognize the host surface, germinates, and then forms a specialized structure called the appressorium to penetrate into the host cell [2,3]. The substance in the conidium, including glycogen and glycerol, is continuously transported into the appressorium and accumulates in it because of its thick melanin layer, which results in tremendous turgor [4]. Due to the turgor pressure, a penetration peg forms to penetrate into the host cuticle. Once entering the host epidermal cells, invasive hyphae (IH) continuously extend and form a series of branches, and then the IH expands from infected cells to adjacent cells [5,6].

Target of rapamycin (TOR) is a conserved serine/threonine (Ser/Thr) protein kinase regulating numerous pathways in cell physiology by sensing environmental conditions [7]. In eukaryotes, TOR can form two kinds of complexes, i.e., TOR complex 1 (TORC1) and TOR complex 2 (TORC2). Among these complexes, TORC1 is sensitive to rapamycin, but TORC2 is not. In animals, TORC1 has been reported to regulate cell growth by maintaining a balance between anabolism and catabolism, such as the production of proteins, lipids, and nucleotides, while TORC2 mainly regulates survival and reproduction [8]. In animals, TORC1 inhibits autophagy by regulating the phosphorylation of ULK1 and Atg13 in the ULK1 complex [9]. In *M. oryzae*, MoTor coordinates with MoSnt2, an epigenetic factor, to regulate autophagy [10]. MoVast1 likely regulates autophagy by promoting MoTor activity [11]. Although studies have shown some regulation of MoTor in *M. oryzae*, knowledge of the TOR pathway is still limited.

Purines, ubiquitous in organisms, are essential for living organisms. As previously reported, purines can act as the building blocks of inheritance (DNA and RNA), energy metabolism (ATP and GTP), signal transduction (cAMP), and be incorporated into coenzymes (NAD+, NADP+ and coenzymes A) [12,13]. Purines can be generated from both the *de novo* purine biosynthesis (DNPB) pathway and the purine salvage pathway. The salvage pathway is usually not enough to meet the demand of purines in most proliferating cells, resulting in activating DNPB. The DNPB pathway is highly conserved and energy intensive. In the DNPB pathway, phosphoribosyl pyrophosphate (PRPP) is converted to inosine monophosphate (IMP) through ten sequential steps. IMP is the first purine nucleotide synthesized and can be converted to adenosine monophosphate (AMP) and guanosine monophosphate (GMP) [14,15]. Six enzymes are involved in the human DNPB pathway. Some of these enzymes are multifunctional: trifunctional GART (TGART) encodes three enzymes, while phosphoribosylaminoimidazole carboxylase (PAICS) and AICAR transformylase/IMP cyclohydrolase (ATIC) are bifunctional and the remaining three enzymes each catalyze a process [12,16]. In yeast, the multifunctional enzymes correspond to different enzymes, and eight enzymes are involved in the DNPB pathway [17]. Extensive studies on the enzymes have revealed that DNPB metabolons called purinosomes can be formed, organizing enzymes in space and time to deliver products economically [18]. In human, diseases caused by disorders of purine metabolism have been reported, which suggests that even partial blocking of purine synthesis can lead to serious consequences [19,20]. Because of their function in cell proliferation, purine antimetabolites, developed as chemotherapeutic drugs, are widely used in the clinical treatment of cancer [21,22].

Glycinamide ribonucleotide transformylase (GART) is the enzyme catalyzing the third step of the DNPB pathway. In the third step, the formyl group of N^10^-formyltetrahydrofolate was transferred to glycinamide ribonucleotide (GAR) to generate formylglycinamide ribonucleotide (FGAR) [23,24]. Previous studies have shown that GART plays crucial roles in various organisms. Deletion of *ADE8* in yeast led to the cessation of cell division, the accumulation of trehalose, and an increase in tolerance to desiccation stress [25]. GART mutant of zebrafish had defects in melanin-derived pigmentation and caused microphthalmia due to abnormal cell cycles [26]. Although many genes in the DNPB pathway have been well characterized, the exact role of GART and many other enzymes involved in DNPB pathway of *M. oryzae* remains unclear. Moreover, whether GART in *M. oryzae* affects other pathways has not been discussed previously. In this study, we focused on DNPB pathway in *M. oryzae* and characterized MoAde8, the homologous protein of GART. MoAde5,7, MoAde6, and MoAde12, homologues of glycinamide ribonucleotide synthase (GARS) and aminoimidazole ribonucleotide synthase (AIRS), formylglycinamidine synthase (FGAMS), and adenylosuccinate synthetase (ADSS) were also identified. Our study showed that the DNPB pathway was involved in many physiological processes of *M. oryzae*, including aerial hyphal growth, conidiation, and pathogenicity. Deletion of MoAde8 increased the sensitivity of the mutant to reactive oxygen species (ROS) and hyperosmolarity. Moreover, our study sheds new light on the regulation of the TOR pathway in *M. oryzae*. Despite its role in regulating the synthesis of nucleotides, MoTor activity can be regulated by the content of purine nucleotides. ∆*Moade8* showed less MoTor activity than Guy11, which was partly due to the blockade of purine nucleotide biosynthesis. The study of the function of MoAde8 provides a new idea for us to understand the pathogenic mechanism of *M. oryzae* and may help us to control rice blast disease.

## 2. Materials and Methods

### 2.1. Strains and Culture Conditions

Guy11 of *M. oryzae*, courtesy of N. J. Talbot from the University of Exeter, was used as the wild type in this article for all the transformation assays. For pathogenesis-related morphology experiments, both the wild type (Guy11) and the transformants (∆*Moade8* and ∆*Moade8*-C) were cultured on complete medium (CM) or minimum medium (MM) agar plates at 25 °C, under a photoperiod of 16 h light and 8 h dark. Adenine (BBI Life Science, Inc., Shanghai, China) was added to CM medium to rescue the morphology of ∆*Moade8* under the same culture conditions. For exogenous stress tests, 0.6 M sodium chloride (NaCl) (Sinopharm Chemical Reagent Co., Ltd., Shanghai, China) and 10 mM hydrogen peroxide (H_2_O_2_) (Sinopharm Chemical Reagent Co., Ltd.) were added into CM plates for hyperosmotic stress and oxidative stress, respectively. All the strains were cultured at 25 °C in the dark. For RNA extraction and protein extraction, mycelial plugs of each strain were harvested from 7-day-old colonies and were transferred into CM liquid media. Then, the contents were placed in a shaker and cultured at 150 rpm under 25 °C for 36 h.

### 2.2. Deletion and Complementation Assay

The deletion vector was constructed following the instructions of the high-throughput gene knockout system as previously described [27]. To construct the deletion vector, 1500 bp sequences were amplified from both upstream and downstream of the target genes. The hygromycin-resistance cassette (HPH) sequence was gained from our laboratory. All the sequences were ligated into the PKO3A vector digested with *Xba*I and *Hin*dIII (Thermo Scientific, Inc., Waltham, MA, USA) using ligase (Transgen, Inc., Beijing, China). The fusion vector was transformed into Guy11 using an *Agrobacterium tumefaciens*-mediated transformation (ATMT) assay to replace the CDS region of *MoADE8* in Guy11 with an HPH sequence. Mutants of other genes were also obtained by this assay, as previously described [28]. To construct the complementation vector, full lengths of the target genes were amplified from the Guy11 genome, and ligated with the PKD5-GFP vector digested with *Bam*HI and *Sma*I (Thermo Scientific, Inc.). The primers used are listed in Appendix A. The vectors were transformed into ∆*Moade8* by ATMT assays to obtain the transformants ∆*Moade8*-C. ∆*Moade8*-C, containing a MoAde8 fused with GFP, was also used to indicate the location of MoAde8.

### 2.3. Fungal Growth, Conidiation and Adenine Treated Assay

To test fungal growth and conidiation, agar plugs were cut from 7-day-old strains of all the transformants and inoculated on CM or MM agar plates. Diameters and conidia were measured at 8 days post-inoculation (dpi). Conidiophores were monitored under a microscope at 24 h post-inoculation (hpi). Exogenous adenine at different concentrations was added to CM to observe its effect on growth and conidiation. Appressoria were induced by inoculating conidia on hydrophobic plastic slides in the dark at 25 °C for 24 h. 

### 2.4. Phenotypic Assays on Rice and Barley Leaves

Phenotypic assays were carried out on barley (*Hordeum vulgare* cv. ZJ-8) and rice (*Oryzae sativa* cv. CO-39) leaves. Both agar plugs and 20 μL of conidial suspension were inoculated as previously described [29]. Disease symptom development was recorded and photographed at 4 dpi. Penetration assays were conducted to observe IH by inoculating conidia suspension on barley leaves. The leaves were collected at 24 hpi and 48 hpi, and were then decolorized with methanol. To test the pathogenicity of ∆*Moade8* on rice, 2.5 mL of conidial suspension (5 × 10^4^ conidia/mL) in 0.2% gelatin of each strain was sprayed on 14-day-old rice seedlings. The lesions were calculated and photographed at 6 dpi. All the assays were repeated three times. 

### 2.5. Fluorescence Observation

To determine the localization of MoAde8, the CDS region of *MoADE8* was amplified and ligated into PKD5-GFP. Then the PKD5-*MoADE8*-GFP fusion plasmid was transferred into ∆*Moade8*. Peroxisomes localization is marked by MoPts1-mCherry [30,31]. Green and red fluorescence were scanned and photographed by fluorescent microscope (Nikon, Inc. Tokyo, Japan).

### 2.6. Western Blot Analysis

For mitogen-activated protein kinase (MAPK) proteins phosphorylation analysis, total protein was extracted using TCA assay and separated by sodium dodecyl sulfate-polyacrylamide gel electrophoresis (SDS-PAGE). To test the phosphorylation of MoPmk1, total proteins were extracted from Guy11 and ∆*Moade8* cultured in yeast-extract glucose (YEG) liquid medium for 36 h at 25 °C and 150 rpm. The phosphorylation level was detected using the phosphor-Pmk1 antibody (Cell Signaling Technology, Inc., Danvers, MA, USA) and Pmk1 antibody (Santa Cruz Biotechnology, Inc., Dallas, TX, USA). For the detection of phosphorylation of MoOsm1, Guy11 and ∆*Moade8* plugs were first cultured in CM liquid medium for 36 h, and then transferred to CM liquid medium containing 0.6 M NaCl for 5, 10, 30, 60, and 90 min. The phosphorylation level was detected using a phosphor-Osm1 antibody (Cell Signaling Technology, Inc.). 

For MoTor kinase activity analysis, the MoRps6-Flag fusion protein was transferred into Guy11 and ∆*Moade8* to replace the original MoRps6. The transformants were cultured in CM liquid medium for 36 h and the total protein was then extracted by TCA assay and separated by phos-tag assay. Phos-tag analysis was conducted using a 10% SDS-polyacrylamide gel added with 5 mM Phos-binding reagent acrylamide (APExBIO, Inc. Houston, USA) and 10 mM MnCl_2_ under the instructions of the manufacturer. 

For autophagy flux detection, Guy11 and ∆*Moade8* containing GFP-MoAtg8 were cultured in CM medium and induced in SD-N liquid medium for 3 h and 6 h. Protein was extracted with extraction buffer and separated by SDS-PAGE [32]. Rapamycin (50 nM) was used to induce autophagy. Anti-GFP antibody was used to detect the levels of GFP-MoAtg8 and GFP (HuaBio, Inc., Hangzhou, China). The ratio of free GFP to the total amount of free GFP and GFP-MoAtg8 fusion protein was calculated.

### 2.7. Quantitative Real-Time Polymerase Chain Reaction (RT-PCR) Assay

Total RNA was extracted from both Guy11 and ∆*Moade8* using Trizol assay. For quantitative RT-PCR (qPCR), the primers used are listed in Appendix A. RNA reverse transcription was conducted using the PrimeScrip^TM^ RT reagent Kit (Takara, Inc, Kyoto, Japan) according to the manufacturer’s instructions. TB Green^®^ Premix EX Taq^TM^ was used as the enzyme and indicator for qPCR amplification. The procedure and relative expression rates were conducted and calculated as previously described [33].

### 2.8. Yeast Two-Hybrid Assay and Co-Immunoprecipitation Assay

The CDS regions were amplified and cloned into the bait vector pGBKT7 (BD) and prey vector pGADT7 (AD). Both the vectors were transformed into yeast strain AH109 to carry out yeast two-hybrid assays. The transformants were first cultured on SD-Leu-Trp synthetic medium and then diluted by water and inoculated on SD-Leu-Trp synthetic medium and SD-Leu-Trp-Ade-His synthetic medium. The results were recorded at 4 dpi. 

For the co-immunoprecipitation (co-IP) assay, MoAde4-3 × Flag was introduced into ∆*Moade8*-C, complemented with MoAde8-GFP. Total protein was extracted and incubated with anti-GFP agarose for 4 h at 4 °C (Smart-Lifesciences, Inc., Changzhou, China). Total protein and protein in the eluate were detected with anti-GFP and anti-Flag antibodies (HuaBio, Inc.) by Western blot.

## 3. Results

### 3.1. Identification of MoAde8 in M. oryzae

Previous studies have already defined rapamycin as a specific inhibitor of mTORC1 that may induce autophagy [11]. To define the critical role of rapamycin in *M. oryzae*, differentially expressed genes were detected using RNA-seq technology. The wild type Guy11 was treated with 30 nM rapamycin for 3 h and the hyphae were collected for further analysis. From the results, we found *MGG_13813*, defined as phosphoribosylglycinamide formyltransferase, was downregulated by a factor of 2.6. According to the National Center for Biotechnology Information (NCBI) website (https://www.ncbi.nlm.nih.gov/ (accessed on 15 March 2021)), we identified it as the homologue of Ade8 in yeast and the C-terminal section of TGART in human and defined it as MoAde8, which also catalyzed the third step of the DNPB pathway in *M. oryzae* [34] (Figure 1). Sequence alignment (Appendix A) and phylogenetic analysis (Appendix A) based on amino acid sequences proved similarity among MoAde8 and GART in *Homo sapiens*, *Neurospora crassa*, *Saccharomyces cerevisiae*, *Fusarium graminearum*, and *Arabidopsis thaliana*, which proved its conservation from fungi to mammals and plants. Conserved domains were predicted by the website Pfam (http://pfam.xfam.org/ (accessed on 11 May 2021)). As shown in Figure 2A, MoAde8 contains a Formyl_trans_N domain family. Similarly, this family also exists in the organisms mentioned above. 

Other proteins involved in the DNPB pathway and AMP biosynthesis in M. oryzae were also identified. These proteins were mentioned as MoAde4 (MGG_04618), MoAde5,7 (MGG_11343), MoAde6 (MGG_1154*1*), MoAde2 (*MGG_01256*), MoAde1 (MGG_12537), MoAde13 (*MGG_03645*), MoAde17 (*MGG_04435*), and MoAde12 (*MGG_17000*) after being compared by the NCBI website (Figure 1).

As proved in previous studies, enzymes in the DNPB pathway well-orchestrated with each other to form purinosomes to make the whole pathway more economical and faster. The product 5-phosphoribosylamine (5-PRA) from the first step has a short half-life under normal physiological conditions. The short-existing state led us to speculate whether there was an interaction between enzymes during product delivery [35]. Then we wondered if an interaction might exist between MoAde8 and other enzymes. Yeast two-hybrid assays were conducted to test this hypothesis (Figure 2B). The results showed that MoAde8 could interact with MoAde4, the enzyme catalyzing the first step in the DNPB pathway. The interaction was also proven by a co-IP assay. The recombinant protein MoAde4-Flag could be pulled down by MoAde8-GFP but could not be pulled down by GFP (Figure 2C). The interaction between MoAde8 and MoAde4 could provide another case for the existence of purinosomes in *M. oryzae*.

### 3.2. MoAde8 Is Essential for Conidiation and Virulence

To define the function of MoAde8 in *M. oryzae*, we knocked out MoAde8 in Guy11. A PCR assay was conducted to detect whether *MoADE8* existed in both the Guy11 and ∆*Moade8* genomes (Appendix A). A complementation assay was carried out and the transformant ∆*Moade8*-C was obtained.

The morphology of ∆*Moade8* was monitored on CM medium plates and MM medium plates, respectively. ∆*Moade8* produced thinner aerial hyphae than Guy11 and ∆*Moade8*-C on CM medium and could not grow on MM medium (Figure 3A). ∆*Moade8* was larger in diameter than Guy11 and ∆*Moade8*-C on CM plates (Figure 3B). Meanwhile, ∆*Moade8* produced more melanin when it was cultured in CM liquid medium for more than 36 h, resulting in a darker color of the liquid medium (Figure 3C). As reported in previous studies, melanin played an essential role in pathogenesis-related processes in *M. oryzae* and was coordinated by three genes, i.e., *MoALB1*, *MoRSY1*, and *MoBUF1* [36]. Total RNA was extracted and the amounts of transcripts of these three genes were also analyzed by the qPCR assay. The primers used are listed in Appendix A. The expression levels of *MoALB1*, *MoRSY1*, and *MoBUF1* were all higher in ∆*Moade8* than in Guy11 (Appendix A).

The function of MoAde8 in conidiation was then evaluated. The Δ*Moade8* mutant was significantly reduced in conidiation and produced less than 200 conidia per culture plate (Figure 3D). Conidiophores in Guy11, ∆*Moade8*, and ∆*Moade8*-C were also induced and photographed. Consistent with the results above, ∆*Moade8* produced almost no conidiophores (Figure 3E).

We then tested whether MoAde8 was important for the pathogenicity of *M. oryzae*. The pathogenicity of ∆*Moade8* was tested on both barley and rice leaves. As shown in Figure 3F,G, mycelial plugs of ∆*Moade8* produced slight lesions, while plugs from Guy11 and ∆*Moade8*-C led to severe, expanded lesions. Taken together, these results indicate that MoAde8 is crucial for the conidiation and virulence of *M. oryzae*, but is not required for substrate hyphal growth. 

### 3.3. Exogenous Adenine Rescues the Pathogenicity-Related Phenotypes of ∆Moade8

Similar to auxotrophic mutants in *M. oryzae* characterized previously [37,38], ∆*Moade8* grew abnormally on MM medium. In yeast, exogenously adding adenine could rescue the phenotype of strains with adenine bradytroph. Adenine could function by being absorbed and metabolized to adenosine diphosphate (ADP) [39]. Exogenous adenine at different concentrations was then added to CM and MM medium to explore whether defects of ∆*Moade8* could be restored. Exogenous adenine could rescue the pathogenicity-related phenotypes of ∆*Moade8* in a dose-dependent manner. Growing on CM supplemented with 0.1 mM, 0.25 mM, 0.5 mM, and 1 mM adenine gradually restored the morphology and conidiation of ∆*Moade8*, and more conidiophores were observed as well (Figure 4A). The conidia produced by ∆*Moade8* growing on CM media containing different concentrations of adenine were calculated, which also showed a dose-dependent form. When the concentration reached 1 mM, ∆*Moade8* was the same as Guy11 in morphology and conidia production, which indicated a full restoration of ∆*Moade8* (Figure 4B). Similar results were obtained when ∆*Moade8* grew on MM medium supplemented with different concentrations of adenine (Appendix A). Hereinafter, conidia from ∆*Moade8* used in the following experiments were collected from CM supplemented with 1 mM adenine.

After being collected and photographed under the microscope, the conidia of ∆*Moade8* were indistinguishable in shape from that of Guy11 (Appendix A). Induced on the hydrophobic surface, no apparent difference was found in appressorium formation rate or appressorium morphology between Guy11 and ∆*Moade8* growing on CM supplemented with 1 mM adenine (Appendix A).

Then, we wondered if the virulence of ∆*Moade8* would be recovered by adding exogenous adenine, and if conidia produced under supplementation with adenine would act normally in pathogenicity. Both the mycelial plugs and the conidial suspensions of Guy11 and ∆*Moade8* were inoculated on detached barley leaves and co-cultivated for 4 d as previously described. The pathogenicity of the mycelial plugs was recovered with increasing adenine concentration, but the pathogenicity of conidia was not recovered by the addition of exogenous adenine to CM medium (Figure 4C,D). 

The rice spraying assay was then performed to attest the attenuated pathogenicity of the conidia of ∆*Moade8* on rice. Conidial suspensions were sprayed onto 14-day-old CO-39 seedlings. The density of the lesion area was analyzed, and for each strain, 15 leaves were measured. As we can see from Figure 4E, ∆*Moade8* produces tiny, restricted lesions while Guy11 and the complemented strain produce dense, severe lesions. Conidia washed from Guy11 growing on CM supplemented with 1 mM adenine were used as the control to illustrate that the addition of adenine did not affect pathogenicity. 

We queried if the compromised pathogenicity of conidia was due to a lack of adenine in the conidial suspension. Then, 1 mM adenine was supplemented into the conidial suspension and the pathogenicity assays were repeated on barley leaves. The pathogenicity was only slightly recovered when 1 mM adenine was added to the conidial suspension (Figure 4F). In summary, exogenous adenine can rescue the morphology and virulence of ∆*Moade8*, especially conidiogenesis, but exogenous adenine cannot completely recover the pathogenicity of the conidia. 

### 3.4. MoAde8 Are Associated with Invasive Growth

To obtain more evidence that deletion of MoAde8 led to defective pathogenicity of *M. oryzae*, penetration assays were performed using the conidia suspension on detached barley leaves. At 24 hpi, Guy11 and ∆*Moade8*-C could successfully penetrate into barley cells, while most of the appressoria of ∆*Moade8* failed to penetrate. Adenine added to conidial suspension could rescue this defect of appressoria. At 48 hpi, the IH of Guy11 and ∆*Moade8*-C showed many branches and could extend to adjacent cells, while the IH of ∆*Moade8* showed few branches and infected one cell. Adenine added to conidial suspension could not resume invasive growth (Figure 5A).

Then, we performed a quantitative analysis of invasive growth. The IH of each strain were classified into three types (type 1, no IH were found; type 2, IH penetrated successfully, and could only infect one cell; type 3, IH could expand to adjacent cells) (Figure 5B). The IH types of all the strains were statistically analyzed at each time point. ∆*Moade8* could be distinguished from Guy11 and ∆Moade8-C in IH types. At 24 hpi, 76.8% of the IH of ∆*Moade8* remained at type 1, while most of that were type 2 in Guy11 and ∆*Moade8*-C. At 48 hpi, most IH were type 3 in Guy11 and ∆*Moade8*-C, while in ∆*Moade8*, 75.7% of the total IH observed were classified into type 2 and only 2.53% were type 3 (Figure 5C). Penetration assays showed that compared to the wild type, IH of ∆*Moade8* had defect in penetration and sprayed more slowly to adjacent cells. Adenine (1 mM) was added to conidial suspension again to attest the role of adenine in invasive growth. We observed that invasive growth was partly recovered at 24 hpi, with 26.8% of the IH belonged to type 1 and 72.1% belonging to type 2, similar to Guy11. However, at 48 hpi, IH could not expand successfully even when adenine was added to conidial suspension, with only 18.3% becoming type 3 and the remaining being type 1 and type 2. 

Previous studies showed a correlation between the Pmk1 pathway and virulence. The Pmk1 pathway is important for appressorium formation, appressorium penetration, and invasive growth in *M. oryzae* [40,41]. We then examined the phosphorylation level of MoPmk1 by Western blot assay, which directly reflected whether the Pmk1 pathway was affected. Compared with Guy11, the phosphorylation level of MoPmk1 in ∆*Moade8* was significantly reduced as we predicted, which might explain the reduced invasive growth of ∆*Moade8* (Figure 5D). We then concluded that MoAde8 was involved in invasive growth.

### 3.5. Deletion of MoADE8 Reduces Tolerance to Hyperosmotic Stress 

During host penetration, fungi need to quickly identify and respond to changes in the microenvironment to penetrate and extend successfully inside host cells, one of which is hyperosmotic stress. Fungi are exposed to increasing osmotic pressure due to the accumulation of degradation products from dead plant cells [42]. Therefore, fungi have evolved a sensitive signaling pathway to deal with external changes. Then, we performed a NaCl stress assay to examine the sensitivity of ∆*Moade8* to hyperosmotic stress. Guy11, ∆*Moade8*, and ∆*Moade8*-C were inoculated on CM containing 0.6 M NaCl and harvested at 8 dpi. As shown in Figure 6A,B, ∆*Moade8* exhibited a higher inhibition rate than Guy11 and ∆*Moade8*-C. Our finding suggests that deletion of MoAde8 leads to increased sensitivity to hyperosmotic stress.

Osm1 MAPK (homology of Hog1 MAPK in yeast) is a pathway functioning in hyperosmotic stress adaptability in *M. oryzae*. MoOsm1 can be rapidly phosphorylated under hyperosmotic stress [43,44]. Thus, we examined the MoOsm1 phosphorylation level in ∆*Moade8* to determine whether the sensitivity of the mutant to hyperosmotic stress was due to the abnormal function of the Osm1 pathway. Both strains were cultured in liquid CM for 36 h and then transferred to fresh liquid CM containing 0.6 M NaCl to induce a hyperosmotic stress response. Then the phosphorylation level of MoOsm1 was detected by Western blot assay. When being exposed to NaCl stress, the phosphorylated MoOsm1 gradually increased in the first 10 min and fell after 30 min. Compared with Guy11, MoOsm1 in ∆*Moade8* phosphorylated normally in the first 10 min but fell faster. After 60 min, phosphorylated MoOsm1 almost disappeared. Furthermore, the level of phosphorylated MoOsm1 of ∆*Moade8* at each time point was lower than the level in the wild type (Figure 6C), indicating decreased activity of the Osm1 MAPK pathway. These findings indicate that MoAde8 is involved in regulating the Osm1 MAPK pathway.

### 3.6. MoAde8 Is Related to Oxidative Stress Response and Co-Localize with Peroxisomes 

Another challenge faced by the fungi in invasive growth is how to cleave ROS. The ROS burst is one of the main immune responses of plants, which can directly inhibit the growth of pathogenic fungi and acts as a signal to induce other immune responses [45,46]. To explore the sensitivity of ∆*Moade8* to oxidative stress, both Guy11 and ∆*Moade8* were inoculated on CM with 10 mM H_2_O_2_. The relative growth rates were compared at 8 dpi. The tolerance of ∆*Moade8* to H_2_O_2_ decreased significantly (Figure 7A,B). 

Elimination of ROS is a major metabolic function of peroxisomes, which is essential for fungal infection [47]. To define the subcellular localization of MoAde8, the MoAde8-GFP fusion expression cassette was transferred into ∆*Moade8*. Being observed in mycelium, MoAde8 dispersed in the cytoplasm (Figure 7C), while in conidia, we observed a clustering of MoAde8-GFP (Figure 7D). Based on the previous conclusions and the punctate distribution, we speculated that the distribution of MoAde8 might be related to peroxisomes. The MoPts1-dsRED fusion protein was transferred into ∆*Moade8*-C as a marker for peroxisomes. Being observed directly, only a few conidia showed co-localization between the two fusion proteins. However, in the germination procedure, being induced in CM liquid medium for 4 h, MoAde8-GFP co-localized with MoPts1-dsRED in 47.4% of the conidia (Figure 7D). Our findings suggest that MoAde8 may cooperate with peroxisomes to regulate the adaptability of *M. oryzae* to oxidative stress.

### 3.7. Deletion of MoAde8 Negatively Regulates TOR Kinase Activity

TOR is an essential kinase involved in a series of anabolic pathways. Once activated, mTORC1 phosphorylates downstream proteins and stimulates the synthesis of proteins and nucleotides, including pyrimidine *de novo* synthesis and the DNPB pathway [48,49]. To investigate whether MoAde8 would in turn affect the activity of MoTor kinase, we detected the level of phosphorylated MoRps6, a protein acting downstream of TORC1 [50]. As shown in Figure 8A, the phosphorylated MoRps6 in ∆*Moade8* decreased significantly. We next explored the reasons for the MoTor inhibition, and we found a slight recovery in mTORC1 activity when ∆*Moade8* was cultured in CM liquid medium containing 1 mM adenine. We speculate that deletion of MoAde8 may inhibit MoTor activity because of the blocked synthesis of purine nucleotides.

TOR regulates the level of autophagy, which has been reported in mammals, plants, and fungi [51,52,53]. Disorders of autophagy in *M. oryzae* have been reported to affect pathogenicity, both decrease and increase of autophagy flux may result in compromised pathogenicity [54,55]. As a response regulated by TOR kinase, the autophagy flux was then examined in the mutant to further confirm the activity of MoTor in ∆*Moade8*. The autophagy level in ∆*Moade8* was slightly higher than that in the wild type when being induced under nitrogen starvation (Figure 8B). Rapamycin was then used to induce autophagy of Guy11 and ∆*Moade8*, respectively. Increased autophagy flux was shown by GFP-MoAtg8 degradation after treatment of rapamycin in ∆*Moade8* (Figure 8C). Based on these results, we queried whether MoAde8 was involved in sensing rapamycin or whether this phenomenon was only caused by the attenuated activity of MoTor. The stress test of rapamycin showed that ∆*Moade8* was not sensitive to rapamycin (Appendix A). Taken together, these results suggest that deletion of MoAde8 results in a compromise in maintaining a stable level of MoTor activity and leads to an upregulated autophagy level.

### 3.8. De Novo Purine Nucleotide Biosynthesis Is Required for the Conidiation and Pathogenicity of M. oryzae

We also tried to knock out other genes functioning in purine nucleotide synthesis to explore the function of the purine nucleotide synthesis pathway. MoAde5,7 is the homologue of GARS and AIRS, catalyzing the second and fifth steps of the DNPB pathway. MoAde6, the homologue of FGAMS, acts downstream of MoAde8 in DNPB pathway. And MoAde12, the homologue of ADSS, is involved in the first step of AMP synthesis from IMP. Using the ATMT method, we obtained a series of mutants of MoAde5,7, MoAde6, and MoAde12 (Appendix A) and phenotypic analyses were conducted. Similarly, the aerial hyphae of these mutants were thinner (Figure 9A). They also showed dramatically decreased conidia and sparse conidiophores. Exogenous adenine was added to CM and MM plates to confirm that these mutants were also auxotrophic. Unsurprisingly, both aerial hyphal growth and conidiation were restored in the case of exogenous addition of adenine (Figure 9A–C and Appendix A). Pathogenetic assays on barley leaves showed that the pathogenicity of *M. oryzae* was also attenuated when these genes were deleted. Similarly, the pathogenicity of mycelial plugs of these mutants could be rescued by additional adenine but the pathogenicity of conidia produced on CM medium containing 1 mM adenine was still lost, similar to ∆*Moade8* (Figure 9D,E). Taken together, these results suggest that blocking purine nucleotide biosynthesis severely interrupts conidiation and pathogenicity in *M. oryzae*.

## 4. Discussion

Purine plays an essential role in cellular metabolic pathways, including energy metabolism, cell signaling, and organismal genetic makeup. Dysregulation of the DNPB pathway leads to severe defects in several organisms, and even partial obstruction of the pathway can lead to serious consequences. Surveying purine biosynthesis can help us identify potential targets in pathogens and the pharmacological inhibition of this pathway can be applied to anti-cancer research [56,57]. TGART functions in the second, third, and fifth steps of the DNPB pathway in mammals, while in yeast, two different proteins are characterized as the homologues of TGART, and are named as Ade5,7 and Ade8, respectively [58]. In our study, we focused on *de novo* purine nucleotide biosynthesis pathway and characterized MoAde8 and MoAde5,7 in *M. oryzae*. Moreover, MoAde6 and MoAde12 were also characterized. 

Proteins involved in nutrient synthesis have been proven to be required for the normal physiological process of *M. oryzae* in previous studies. Deletion of isopropylmalate isomerase MoLeu1 resulted in leucine auxotroph and defects in vegetative growth and pathogenicity [59]. Disruption of MoArg1, MoArg5,6, and MoArg7 blocked arginine synthesis and affected growth, conidiation, and sexual reproduction of *M. oryzae* [60]. The DNPB pathway has been proven to be essential for the growth of *M. oryzae*. IH of ∆*Moade1* could not grow normally in rice cells [61]. We knocked out MoAde8 and obtained similar results. ∆*Moade8* displayed adenine auxotroph and was unable to grow on MM medium. In addition, ∆*Moade8* had severe defects in infection-related morphology and pathogenicity. Knockout mutants of MoAde5,7, MoAde6, and MoAde12 showed the similar results, suggesting that *de novo* purine nucleotide biosynthesis pathway was required for conidiogenesis and pathogenicity. A recent study also showed that deletion of MoAde12 attenuated virulence in *M. oryzae*. Similar to ∆*Moade8*, ∆*Moade12* is sensitive to osmotic stress and oxidative stress. Ade12 and other proteins involved in *de novo* IMP, AMP, and GMP are also important for vegetative growth, sexual reproduction, and virulence in *F. graminearum* [62,63]. Whether purine metabolism is required for sexual reproduction in *M. oryzae* needs to be further confirmed.

Similar to other nutrient synthesis genes, exogenous adenine could rescue the defects of the mutants, but the pathogenicity of conidia could not be recovered, which caught our attention. We speculated that MoAde8 was involved in regulating other pathways. Disturbing the cAMP-PKA and Pmk1 pathways in ∆*Motea1* resulted in obstructed appressorium formation and virulence [64]. ∆*Moade8* was able to form normal, melanized appressoria without additional adenine in conidia suspension and exhibited normal turgor pressure (Appendix A). Based on our penetration assays, conidia of ∆*Moade8* had defects in both penetration and invasive growth and only the penetration ability could be recovered by additional adenine in conidia suspension. The attenuated level of phosphorylated Pmk1 MAPK in ∆*Moade8* could explain the reduced invasive ability. Moreover, sensitivity to hyperosmotic stress and oxidative stress may also result in defective invasive growth. In a previous study, GART was reported to mediate cellular apoptosis via the MEKK3-MKK3-p38 MAPK pathway, the homologue of the Osm1 MAPK pathway [65]. Our research also showed that MoAde8 is involved in Osm1 MAPK regulation when being exposed to hyperosmotic stress.

In *M. oryzae*, pathogenesis is a complicated process including development of conidia and appressoria, penetration, and IH growth. Melanin has been reported as a key factor for penetration; deletion of either *MoALB1*, *MoBUF1*, or *MoRSY1* prevented the mutants from forming penetration pegs and resulted in loss of pathogenicity [36]. In our study, we found that deletion of *MoADE8* resulted in increased hyphal melanization, while the pathogenicity of ∆*Moade8* was decreased. These results were similar to the results of MoOpy2 and MoCnf1 in previous studies [27,54]. Hyphae and appressoria melanization are regulated by different mechanisms. The expression of melanin biosynthesis genes in hyphae is mainly regulated by the transcription factor Pig1. In contrast, the expression of melanin biosynthesis genes in appressoria is regulated mainly by the transcription factor Vrf1 [27,66].

In addition to the function of MoAde8 on pathogenesis-related morphology, we tried to find the relationship between the DNPB proteins. “Metabolon” was first coined in humans to define a supramolecular complex of a series of sequential metabolic enzymes, providing channels for substrate transportation and regulating the pathway flux [67]. Numerous purine nucleotides need to be synthesized in the process of life activities of normal living organisms, especially in vigorously dividing cells. Co-localization between DNPB enzymes was discovered and defined as purinosomes. Purinosome formation is a dynamic, reversible process that can be regulated by exogenous purine content [68]. In yeast, DNPB enzymes formed condensates, but the mechanism seemed to be different [69]. Based on the concept of purinosomes, we explored whether there was any interaction between the enzymes in the DNPB pathway in *M. oryzae*. MoAde8 was found to physically interact with MoAde4, which catalyzed the first step of DNPB. The physical interaction between PPAT and GART has not been found to date. The interaction between MoAde8 and MoAde4 might provide us with evidence of a cohesive state between the DNPB enzymes.

In previous studies, mTORC1 was proven to stimulate *de novo* purine synthesis by regulating the mitochondrial tetrahydrofolate cycle at the transcriptional level. Treatment with the mTORC1 inhibitor rapamycin decreased purine biosynthesis flux [49]. Another study showed that treated with AG2037, an inhibitor of GARFT (the enzyme catalyzing the third step of the DNPB in mammals), resulted in the inhibition of mTORC1 activity [70]. In our study, MoTor also showed less activity in ∆*Moade8* than in Guy11, which could be restored by exogenous adenine. As a result of the decreased MoTor activity, autophagy flux was lifted after being induced by nitrogen starvation. Our work suggested that the DNPB pathway could in turn regulate MoTor activity partly by the synthesis of purine nucleotides. We found that exogenous adenine could not completely recover the activity of mTORC1 in *M. oryzae*, and there appeared to be some mechanisms between DNPB and MoTor. Interestingly, an interaction was found between MoAde8-interacting protein MoAde4 and the rapamycin-binding protein MoFpr1. Rapamycin can form a complex with 12-kDa FK506-binding protein (FKBP12), and the complex then binds with mTORC1 and acts as the inhibitor of mTORC1 [71]. In *M. oryzae*, FKBP12 was mentioned as MoFpr1. The interaction between MoAde4 and MoFpr1 was proven by both yeast two-hybrid assays and co-IP assays (Appendix A). However, further study on the sensitivity of ∆*Moade4* to rapamycin also showed no significance when compared with Guy11 (Appendix A). Although this interaction did not result in the sensitivity of ∆*Moade4* or ∆*Moade8* to rapamycin, it still suggested a potential relationship between the DNPB and TOR pathway, but the mechanisms underlying these two pathways remain a mystery.

In summary, purine nucleotide biosynthesis is required for conidiation and pathogenesis in *M. oryzae*. Deletion of MoAde8 results in attenuated invasive growth in barley cells, and increases the sensitivity to hyperosmotic and oxidative stress. In addition, MoAde8 is involved in regulating MoTor activity. The *de novo* purine nucleotide biosynthesis pathway, serving as a potential target for inhibiting the growth of *M. oryzae*, may also contribute to the control of rice blast disease.

## Figures and Tables

**Figure 1 jof-08-00915-f001:**
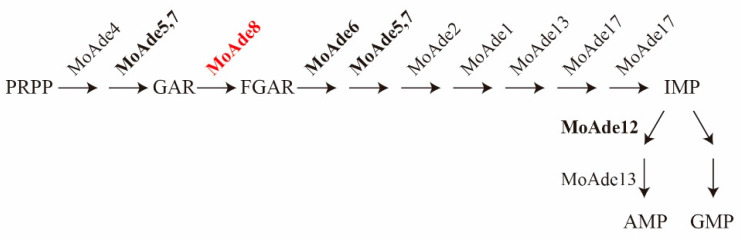
Schematic of the purine nucleotide *de novo* biosynthesis pathway. The *de novo* purine biosynthesis (DNPB) pathway contains 10 steps from PRPP to IMP catalyzed by six enzymes in humans. In *M. oryzae*, the proteins corresponding to these six enzymes are indicated in the figure. Abbreviations: phosphoribosyl pyrophosphate (PRPP), glycinamide ribonucleotide (GAR), formylglycinamide ribonucleotide (FGAR), inosine monophosphate (IMP), adenosine monophosphate (AMP), guanosine monophosphate (GMP).

**Figure 2 jof-08-00915-f002:**
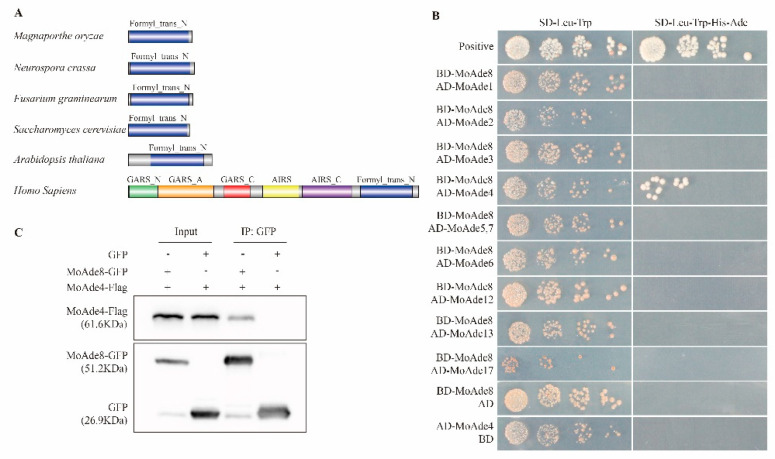
Identification of MoAde8 in *M. oryzae*. (**A**) Schematic diagram of GART homologue proteins in *Neurospora crassa*, *Saccharomyces cerevisiae*, *Fusarium graminearum*, *Arabidopsis thaliana* and *Homo sapiens*. Conserved domains were predicted by Pfam (https://pfam.xfam.org/ (accessed on 11 May 2021)) and drawn by the IBS website (http://ibs.biocuckoo.org/ (accessed on 11 November 2021)). *Neurospora crassa* (XP_963963.1), *Saccharomyces cerevisiae* (NP_010696.3), *Fusarium graminearum* (XP_388605.1), *Arabidopsis thaliana* (NP_174407.1), *Homo sapiens* (NP_000810.1) and *M. oryzae* (XP_003710972.1). (**B**) Yeast two-hybrid assays were conducted to examine the interaction between MoAde8 and other enzymes involved in DNPB pathway. pGBKT7-53 and pGADT7-T were used as the positive controls. (**C**) A co-IP assay was conducted on the strain co-expressed the GFP-fused MoAde8 and 3 × FLAG-fused MoAde4 to verify the interaction of MoAde8 and MoAde4 in vivo. The strain co-expressed GFP and 3 × FLAG-fused MoAde4 was used as the negative control.

**Figure 3 jof-08-00915-f003:**
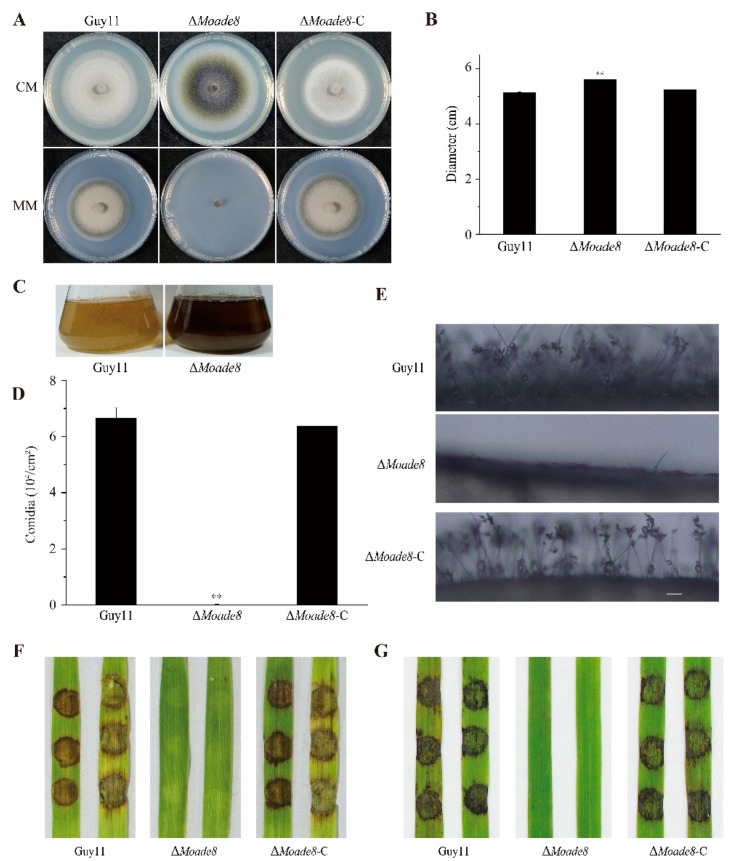
MoAde8 is required for growth, conidiation, and virulence. (**A**) Hyphae growth of Guy11, ∆*Moade8*, and ∆*Moade8*-C on CM and MM medium plates for 8 d. (**B**) The diameters of the colonies grown on the CM medium were measured at 8 days post-inoculation (dpi). Data were analyzed by unpaired two-tailed Student’s *t*-test. Asterisks are used to mark significant differences (** *p* < 0.01). (**C**) Deletion of MoAde8 led to increased melanin synthesis. Mycelium plugs of Guy11 and ∆*Moade8* were cultured in CM liquid medium for 40 h. (**D**) Conidia were harvested from 8-day-old colonies and were analyzed by unpaired two-tailed Student’s *t*-test. Asterisks are used to mark significant differences (** *p* < 0.01). (**E**) Conidiophores of Guy11, Δ*Moade8*, and ∆*Moade8*-C were induced in an incubator at 25 °C for 24 h and observed under an optical microscope. Scale bar, 50 μm. (**F**,**G**) Mycelium plugs of Guy11, Δ*Moade8*, and ∆*Moade8*-C were inoculated on detached barley (**F**) and rice (**G**) leaves. Photographs were taken at 4 dpi.

**Figure 4 jof-08-00915-f004:**
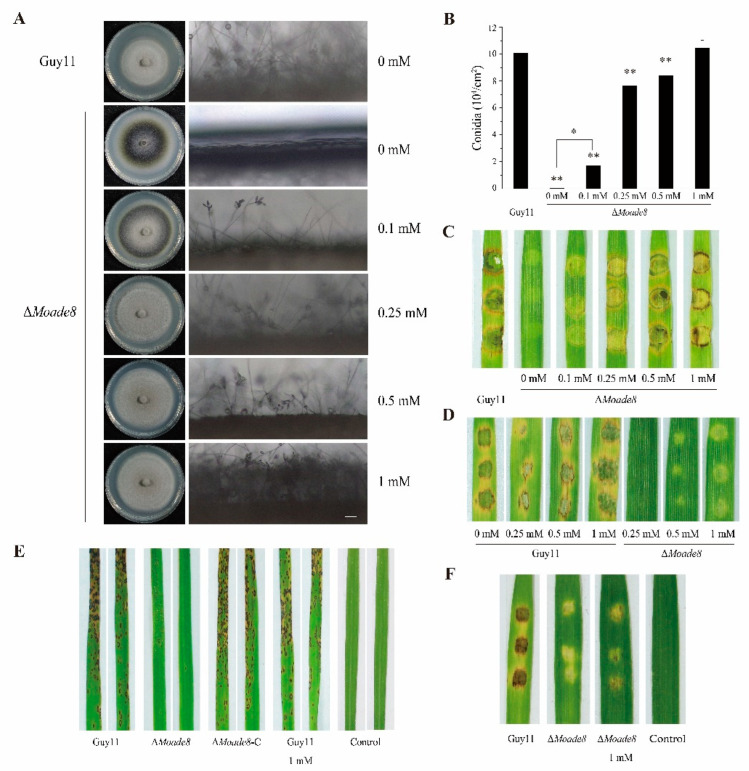
Exogenous adenine rescues the defective pathogenicity-related morphology of ∆*Moade8*. (**A**) Growth and conidiophore induction of ∆*Moade8* on CM supplemented with 0 mM, 0.1 mM, 0.25 mM, 0.5 mM, and 1 mM adenine. Photographs of conidiophores were taken at 24 h post-inoculation (hpi). (**B**) Conidia were harvested from 8-day-old colonies grown on CM supplemented with 0 mM, 0.1 mM, 0.25 mM, 0.5 mM, and 1 mM adenine. Data were analyzed by unpaired two-tailed Student’s *t*-test. Asterisks are used to mark significant differences (* *p* < 0.05, ** *p* < 0.01). (**C**) Mycelium plugs of Guy11 and Δ*Moade8* grown on CM supplemented with adenine were inoculated on detached barley leaves for 4 d. (**D**) Conidial suspensions (5 × 10^4^ conidia/mL) of Guy11 and ∆*Moade8* were inoculated on barley leaves for 4 d. (**E**) Rice seedlings (14-day-old) were sprayed with 5 × 10^4^ conidia/mL conidial suspension from Guy11, ∆*Moade8*, and ∆*Moade8*-C, respectively. Guy11 growing on CM containing 1 mM adenine was used as the control. The sprayed rice seedlings were cultured in incubator for 6 d. (**F**) 5 × 10^4^ conidia/mL conidial suspensions of Guy11, ∆*Moade8* and ∆*Moade8* conidia suspension containing 1 mM adenine was inoculated on barley leaves for 4 d.

**Figure 5 jof-08-00915-f005:**
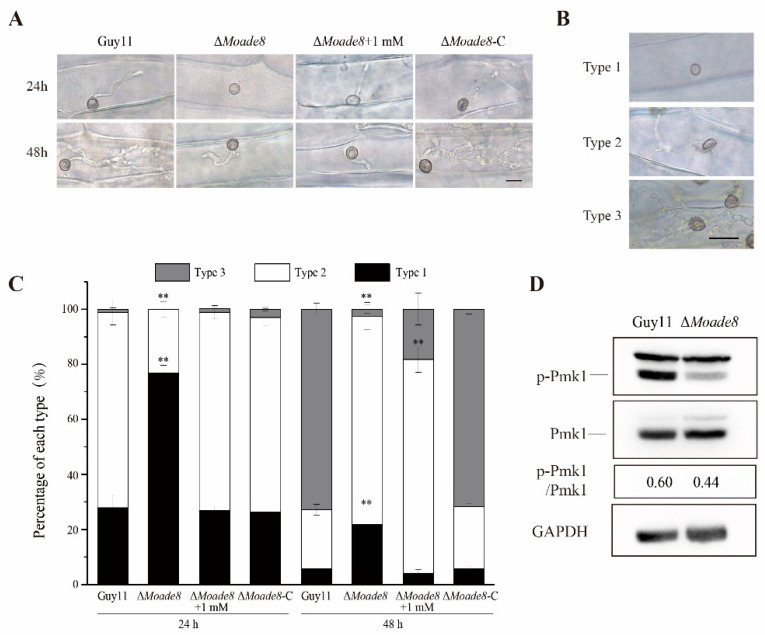
MoAde8 is involved in invasive growth. (**A**) The invasive hyphae (IH) of each strain were photographed at 24 hpi and 48 hpi. Δ*Moade8* + 1 mM: 1 mM adenine was added to the conidial suspension of Δ*Moade8*. (**B**) IH were divided into three types depending on whether IH expanded to adjacent cells. Scale bar, 20 μm. (**C**) IH were observed at 24 hpi and 48 hpi. For each replicate, 100 appressoria were counted. Asterisks are used to mark significant differences (** *p* < 0.01). (**D**) Level of phosphorylated Pmk1 in Guy11 and ∆*Moade8*. GAPDH was detected as the control.

**Figure 6 jof-08-00915-f006:**
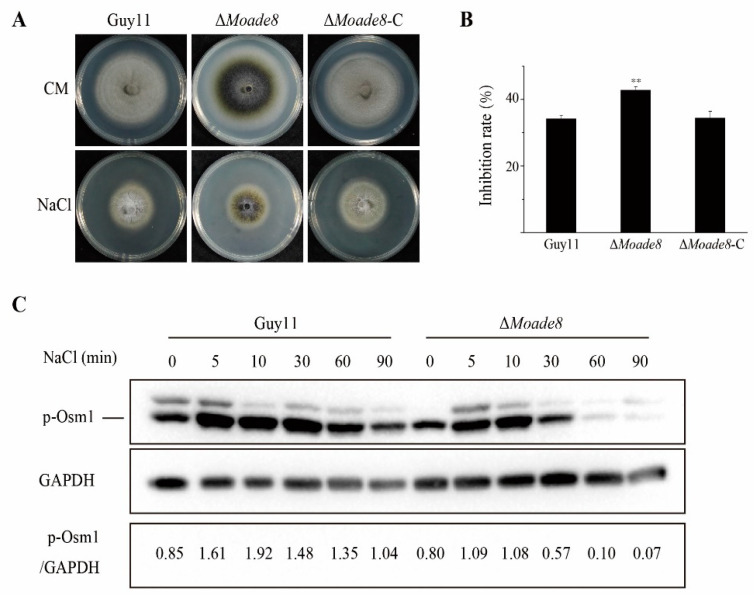
MoAde8 is involved in regulating the Osm1 MAPK pathway. (**A**) Growth of Guy11, ∆*Moade8*, and ∆*Moade8*-C on CM and CM supplemented with 0.6 M NaCl for 8 d. (**B**) Inhibition rates of Guy11, ∆*Moade8*, and ∆*Moade8*-C were calculated when being exposed to NaCl. Asterisks are used to mark significant differences (** *p* < 0.01). (**C**) Phosphorylation level of Osm1 in Guy11 and ∆*Moade8* was detected after induction by liquid CM medium containing 0.6 M NaCl at 0, 5, 10, 30, 60, and 90 min. GAPDH was detected as the control.

**Figure 7 jof-08-00915-f007:**
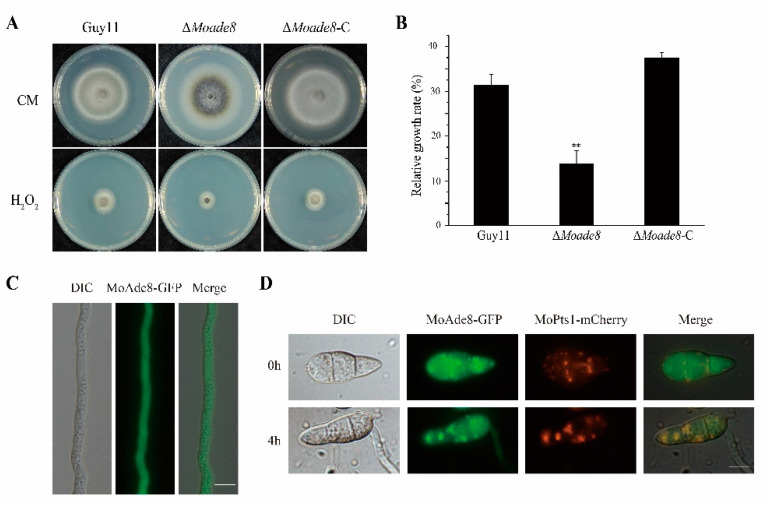
MoAde8 is related to the oxidative stress response. (**A**) Growth of Guy11, ∆*Moade8*, and ∆*Moade8*-C on CM and CM supplemented with 10 mM H_2_O_2_ for 8 d. (**B**) Relative growth rates of Guy11, ∆*Moade8*, and ∆*Moade8*-C were calculated when being exposed to H_2_O_2_. Asterisks are used to mark significant differences (** *p* < 0.01). (**C**) Localization of MoAde8-GFP in the mycelium of *M. oryzae*. Scale bar, 10 μm. (**D**) Co-localization between MoAde8 and peroxisomes. Epifluorescence and DIC images of ∆*Moade8*-C conidia expressing MoPTS1-mCherry were taken before and after a 4 h induction in CM liquid medium. Scale bar, 10 μm.

**Figure 8 jof-08-00915-f008:**
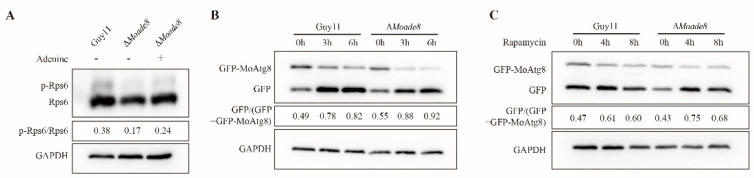
Deletion of MoAde8 negatively regulates TOR kinase activity. (**A**) Phosphorylation levels of MoRps6 were detected by phos-tag assay in Guy11 and ∆*Moade8*. Strains were cultured in CM liquid medium or CM liquid medium supplemented with 1 mM adenine for 36 h. GAPDH was detected as the control. (**B**) Autophagy intensity was detected by Western blot assay in Guy11 and ∆*Moade8*. GFP-MoAtg8 and free GFP were detected by anti-GFP antibody. The ratio of free GFP to the sum of GFP and GFP-MoAtg8 was calculated. GAPDH was detected as the control. (**C**) Rapamycin (50 nM) was used to induce autophagy in Guy11 and ∆*Moade8*. Strains were first cultured in CM for 36 h, and then the mycelia were collected and transferred into CM containing 50 nM rapamycin and cultured for 4 h and 8 h. Degradation of GFP-MoAtg8 was detected by Western blot assay. The ratio of free GFP to the sum of GFP and GFP-MoAtg8 was calculated. GAPDH was detected as the control.

**Figure 9 jof-08-00915-f009:**
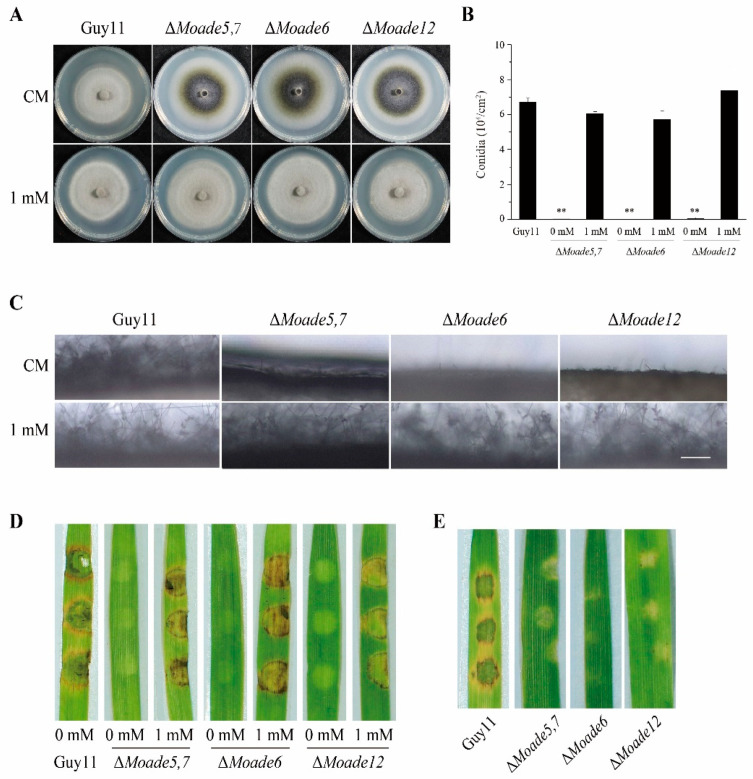
The DNPB pathway is required for conidiation and pathogenicity. (**A**) Hyphae growth of Guy11, ∆*Moade5,7*, ∆*Moade6*, and ∆*Moade12* on CM and CM supplemented with 1 mM adenine for 8 d. (**B**) Conidia were harvested from 8-day-old colonies grown on CM supplemented with 1 mM adenine. Data were analyzed by unpaired two-tailed Student’s *t*-test. Asterisks are used to mark the significant differences (** *p* < 0.01). (**C**) Conidiophores of Guy11, Δ*Moade5,7*, ∆*Moade6* and Δ*Moade12* growing on CM and CM containing 1 mM adenine were induced in an incubator at 25 °C for 24 h. Scale bar, 50 μm. (**D**) Mycelium plugs of Guy11 and different mutants grown on CM supplemented with adenine were inoculated on detached barley leaves for 4 d. (**E**) 5 × 10^4^ conidia/mL conidial suspensions of Guy11 and DNPB pathway mutants were inoculated on barley leaves for 4 d.

## Data Availability

The data presented in this study are available in this published article or its Appendix A.

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
