# Peer review of "De Novo Purine Nucleotide Biosynthesis Pathway Is Required for Development and Pathogenicity in Magnaporthe oryzae"

_jof, 2022, doi:10.3390/jof8090915_

Round 1
Reviewer 1 Report
The paper entitled "De novo Purine Nucleotide Biosynthesis Pathway Is Required 2 for Development and Pathogenicity in Magnaporthe oryzae", has investigated Purine Nucleotide Biosynthesis Pathway Is Required for Development and Pathogenicity in Magnaporthe oryzae, as well as the role the TOR pathway plays.
Although the work is of great scientific importance, it is important that the aims and materials and method sections of the manuscript are more clearly explained.
For example:
1. The authors needs to explain in more detail how the study will help to increase our knowledge about the TOR pathways (TORC1/TORC2) in the aims.
2. More information should be included about the strains used in the study (perhaps references?).
3. The authors need to ensure that all suppliers information are included (for example, hydrogen peroxide (line 119).
4. The authors should explain more clearly the different culture conditions used in this study.
5. RNA work should have its own heading and explained in detail.
Author Response
Major Comments: Although the work is of great scientific importance, it is important that the aims and materials and method sections of the manuscript are more clearly explained.
Responses: Thank you for pointing out the inadequacies of this manuscript. Based on your comments, we have revised the manuscript and further clarified the aim and add more detail the Materials and methods. We hope that it has become much better. What has been changed are listed as follows.
1. The authors needs to explain in more detail how the study will help to increase our knowledge about the TOR pathways (TORC1/TORC2) in the aims.
Response: Thank you for your constructive comment. In M. oryzae, knowledge of Tor signaling is still limited. There’s only one Tor protein in M. oryzae. In our research, MoRps6, downstream of mTORC1 in mammalian cells was used to measure MoTor activity. Deletion of MoAde8 led to decreased MoTor activity. We observed that MoTor activity is partly recovered when adenine was added. We tried to find more association between MoAde8 and protein involved in MoTor pathway, but failed. What we can confirm is that the effect of ade8 on the Tor pathway does not directly act on MoTor complex proteins, such as MoLst8. Yeast two-hybrid assays were used to test the interactions but no exciting results were found. Interestingly, the content of purine nucleotide can affect the activity of MoTor kinase. Details added can be seen at Line 55-58, 105, 118-121.
2. More information should be included about the strains used in the study (perhaps references?).
Response: Thanks for your good suggestion. Related content has been added in Materials and Methods at Line 126.
3. The authors need to ensure that all suppliers information are included (for example, hydrogen peroxide (line 119).
Response: Thanks for your good suggestion. Suppliers information of hydrogen peroxide, sodium chloride, adenine and restriction endonuclease have been added in Materials and Methods. Please see them at Line 130-135, 146 and 152.
4. The authors should explain more clearly the different culture conditions used in this study.
Response: Thanks for your suggestion. More details about the different culture conditions was added in section 2.1. (Line 127-135) We have clearly described the different culture conditions of different experiments and added the culture conditions of exogenous pharmacodynamic stress.
5. RNA work should have its own heading and explained in detail.
Response: Thanks for your suggestion. According to your kind suggestion, we agree it’s not appropriate to put RNA work together with PCR work. Now RNA work is presented in Figure S3 now, a new title is given (Line 645) and more detail are given in the text (Line 285-286 and Figure S3). In our research, RNA work was conducted to reflect the increased expression level of three genes involving in melanin production. According to our results, elevated melanin content in the mutant is not the main cause of the loss of pathogenicity, just a phenotype of the mutant during its growth. So the results of qPCR were only put in supplementary Figure S1.
As for the English writing of the manuscript, we have invited native English speaker to revise this manuscript conscientiously. Grammatical errors and sentences led to poor readability have been revised. Thanks again for your comments!
Reviewer 2 Report
Dear colleagues.
There are several questions.
Figure 6 Line 392-398.
Osmotic stress was created using NaCl. were there any attempts to create such stress using? for example, mannitol?
Figure 7. Line 421-426.
10 mm H2O2 is a sufficiently high concentration of hydrogen peroxide. Have the authors tried to see at what rate hydrogen peroxide is destroyed within 8 days?
Have you watched the germination of conidia in the presence of peroxide?
What enzymes, being in peroxisomes, in your opinion, can play a role in detoxification of ROS?
«Hyphae and appressoria melanization are regulated by different mechanisms» - Line 551
Does this affect the structure of melanin?
Author Response
1. Osmotic stress was created using NaCl. were there any attempts to create such stress using? for example, mannitol?
Response: Thank you for your question. Osmotic stress was also created by 1 M sorbitol, another kind of polyols, which provides nonionic osmotic stress similar as mannitol. (Mannitol is not used.) The diameter of Guy11 and ∆Moade8 showed no significant difference when being exposed to sorbitol. So, the data are not shown in our manuscript. We speculated that the sensing mechanism was different when being exposed to sorbitol and NaCl, possibly because salt as NaCl provides ionic stress while sorbitol is nonionic. A study of our lab showed that ΔMohse1 and ΔMovps27 are also sensitive to NaCl but are not sensitive to sorbitol [1]. Another study in Metarhizium acridum may help prove our thoughts. MaNmrA knockout mutant is only sensitive to sorbitol but is not sensitive to NaCl [2]. So our results showed that ΔMoade8 is only sensitive to ionic stress.
2. 10 mm H2O2 is a sufficiently high concentration of hydrogen peroxide. Have the authors tried to see at what rate hydrogen peroxide is destroyed within 8 days? Have you watched the germination of conidia in the presence of peroxide?
Response: Thank you for your constructive suggestion. H2O2 is easily decomposed. In our stress experiment, all the strains were cultured in dark to ensure minimal H2O2 decomposition. When 5 mM H2O2 was used, Guy11 is not that sensitive to it [3]. In order to have a better experimental result, 10 mM concentration is determined based on previous research in our laboratory [4,5].
Germination rate of conidia from Guy11 and ∆Moade8 have been measured according to your suggestion. When exposed to 10 mM H2O2, 42.5% of the conidia of Guy11 could germinate, and 44.7% of ∆Moade8 could germinate successfully at 2dpi. However, the germinated conidia of WT or mutant could not differentiate to appressoria at 24dpi. Additionally, we observed that conidia adhesion ability of ∆Moade8 decreased when being exposed to exogenous H2O2.
3. What enzymes, being in peroxisomes, in your opinion, can play a role in detoxification of ROS?
Response: Thank you for your question. Catalase, a kind of peroxisomal matrix proteins, play major roles in protecting organisms against peroxide stress. It can be imported in a peroxisome-targeting signal type-1 (PTS1) dependent manner. CPXB has been identified as a catalase in M. oryzae, and the knockout mutant showed sensitiveness to H2O2[6].
4. «Hyphae and appressoria melanization are regulated by different mechanisms» - Line 551. Does this affect the structure of melanin?
Response: Thank you for your question. 1,8-dihydroxynaphthalene (DHN) melanin in M. oryzae is formed by the polyketide synthase pathway. M. oryzae produces melanin during vegetative growth, forming melanized colonies on CM media. Anyone of MoAlb1, MoBuf1 and MoRsy1 being deleted causes light color in both hypha, conidia and appressoria melanin production. According to previous studies of our lab, the transcription factor Pig1 regulate melanin biosynthesis genes in hyphae and expression of melanin biosynthesis genes in appressoria is regulated by the transcription Vrf1. Although they were regulated by different mechanism, melanin is produced by polyketide synthase pathway and same product is obtained.
Special thanks for your comments!
References:
- Sun, L.X.; Qian, H.; Liu, M.Y.; Wu, M.H.; Wei, Y.Y.; Zhu, X.M.; Lu, J.P.; Lin, F.C.; Liu, X.H. Endosomal sorting complexes required for transport-0 (ESCRT-0) are essential for fungal development, pathogenicity, autophagy and ER-phagy in Magnaporthe oryzae. Environ Microbiol 2022, 24, 1076-1092.
- Li, C.; Zhang, Q.; Xia, Y.; Jin, K. MaNmrA, a Negative Transcription Regulator in Nitrogen Catabolite Repression Pathway, Contributes to Nutrient Utilization, Stress Resistance, and Virulence in Entomopathogenic Fungus Metarhizium acridum. Biology 2021, 10, 1167.
- Wei, Y.Y.; Liang, S.; Zhang, Y.R.; Lu, J.P.; Lin, F.C.; Liu, X.H. MoSec61β, the beta subunit of Sec61, is involved in fungal development and pathogenicity, plant immunity, and ER-phagy in Magnaporthe oryzae. Virulence 2020 11(1):1685-1700.
- Wu, M.-H.; Huang, L.-Y.; Sun, L.-X.; Qian, H.; Wei, Y.-Y.; Liang, S.; Zhu, X.-M.; Li, L.; Lu, J.-P.; Lin, F.-C.; Liu, X.-H. A Putative D-Arabinono-1,4-lactone Oxidase, MoAlo1, Is Required for Fungal Growth, Conidiogenesis, and Pathogenicity in Magnaporthe oryzae. J. Fungi 2022, 8, 72.
- Sun, L.X.; Qian, H.; Wu, M.H.; Zhao, W.H.; Liu, M.Y.; Wei, Y.Y.; Zhu, X.M.; Li, L.; Lu, J.P.; Lin, F.C.; Liu, X.H. A Subunit of ESCRT-III, MoIst1, Is Involved in Fungal Development, Pathogenicity, and Autophagy in Magnaporthe oryzae. Front Plant Sci. 2022, 7, 13:845139.
- Tanabe, S.; Ishii-Minami, N.; Saitoh, K.; Otake, Y.; Kaku, H.; Shibuya, N.; Nishizawa, Y.; Minami, E. The Role of Catalase-Peroxidase Secreted by Magnaporthe oryzae During Early Infection of Rice Cells. Molecular plant-microbe interactions 2011, 24, 163-171.
Reviewer 3 Report
In this study, the authors generated and characterized the MoADE8 deletion mutants in Magnaporthe oryzae. Deletion of MoADE8 resulted in pleiotropic defects in growth, conidiation, and pathogenesis. They showed the MoAde8 interacted with other components of the DNPB pathway, indicating they may function together as the purinosome in M. oryzae. The Moade8 mutant was defective in TOR signaling, autophagy, and responses to oxidative and osmotic stresses. Furthermore, the authors showed that deletion of other genes involved in purine metabolism, including ADE12 and ADE6 orthologs, had similar effects as MoADE8 deletion.
Overall, the authors presented sufficient data to support their conclusions (need rewording for some of their conclusions). The manuscript was well organized. Although the importance of purine metabolism has been documented in several fungal pathogens, the authors thoroughly characterized the phenotypes of the Moade8 mutant and connected the function of MoADE8 with autophagy and TOR signaling.
One major concern is that the authors overstated the function of MoADE8 throughout the entire manuscript. Based on data presented in this manuscript, MoADE8 is important but not essential (such as line 248) or required (such as ling 477) for hyphal growth, conidiation, etc. The entire manuscript should be reworded for the importance of MoADE8.
My other concern is the English. Although it is comprehensible, this manuscript will benefit from a careful revision, possibly with assistance of somebody with English as the native language. For example, line 242, changing to: ‘yeast two-hybrid assays were conducted to test this hypothesis’. The assay or assays – follow the same rule for other descriptions related to the word ‘assay’. Also, not appropriate to describe ‘confirm our thoughts. ….
Another example, line 276-277. This sentence is OK but just not the right way to describe this observation. Rewording will be helpful. ‘Quantitative analysis showed that’ -this phrase is OK but not necessary and not consistent with the second half of this sentence. The ΔMoade8 mutant was significantly reduced in conidiation and produced less than 100? conidia per culture plate (Figure 3D). Few?
There is a recent report on differences in purine metabolism during saprophytic and infectious growth in Fusarium graminearum. The authors may want to discuss their results with this publication in F. graminearum. It is likely that M. oryzae may have similar differences.
Author Response
1. One major concern is that the authors overstated the function of MoADE8 throughout the entire manuscript.
Response: Thank you for pointing out this mistake! The importance of MoAde8 has been reworded in this manuscript. According to our results, MoAde8 is required for aerial hyphae growth, conidiation, and virulence. And other genes involved in DNPB pathway showed similar results. They produced almost no conidia (Figure 3D). (Please see them at Line 112, 260, 297)
2. My other concern is the English. Although it is comprehensible, this manuscript will benefit from a careful revision, possibly with assistance of somebody with English as the native language.
Response: Thank you for your suggestion! I feel really sorry for my poor writing.
We have invited native English speaker to revise this manuscript conscientiously. Grammatical errors and sentences led to poor readability have been revised. We sincerely hope that the level of the language has been improved. Changes have been made in the places pointed out. Misusage of “assay” is all corrected. (Line 265, 302)
Moreover, thanks for your kind suggestion, we’ve read the corresponding article in Fusarium graminearum. Details are added in discussion now. (Line 568-571).